# 4D MRI: Robust sorting of free breathing MRI slices for use in interventional settings

**Gino Gulamhussene**[1], **Fabian Joeres**[1☯], **Marko Rak**[1☯], **Maciej Pech**[2], **Christian Hansen**[1] *

**1** Faculty of Computer Science, Otto-von-Guericke University Magdeburg, Magdeburg, Germany, **2** Clinic for Radiology and Nuclear Medicine, University Hospital Magdeburg, Magdeburg, Germany

☯ These authors contributed equally to this work.
* hansen@isg.cs.uni-magdeburg.de

**Data Availability Statement:** All Dicom files are available from the OPEN SCIENCE Repository for Research Data and Publications of OVGU database (accession url https://doi.org/10.24352/UB.OVGU-2019-093).

## Abstract

### Purpose

We aim to develop a robust 4D MRI method for large FOVs enabling the extraction of irregular respiratory motion that is readily usable with all MRI machines and thus applicable to support a wide range of interventional settings.

### Method

We propose a 4D MRI reconstruction method to capture an arbitrary number of breathing states. It uses template updates in navigator slices and search regions for fast and robust vessel cross-section tracking. It captures FOVs of 255 mm x 320 mm x 228 mm at a spatial resolution of 1.82 mm x 1.82 mm x 4mm and temporal resolution of 200ms. A total of 37 4D MRIs of 13 healthy subjects were reconstructed to validate the method. A quantitative evaluation of the reconstruction rate and speed of both the new and baseline method was performed. Additionally, a study with ten radiologists was conducted to assess the subjective reconstruction quality of both methods.

### Results

Our results indicate improved mean reconstruction rates compared to the baseline method (79.4% vs. 45.5%) and improved mean reconstruction times (24s vs. 73s) per subject. Interventional radiologists perceive the reconstruction quality of our method as higher compared to the baseline (262.5 points vs. 217.5 points, p = 0.02).

### Conclusions

Template updates are an effective and efficient way to increase 4D MRI reconstruction rates and to achieve better reconstruction quality. Search regions reduce reconstruction time. These improvements increase the applicability of 4D MRI as a base for seamless support of interventional image guidance in percutaneous interventions.

**Funding:** This study was supported by funding from the Investitionsbank Sachsen-Anhalt (https://www.ib-sachsen-anhalt.de/) to GG with the grant number 1704/00038. The funder had no role in the study design, data collection and analysis, decision to publish, or preparation of this manuscript.

**Competing interests:** The authors have declared that no competing interests exist.

## Introduction

During the last decade, 4D MRI has gained considerable interest in research, because it promises access to information on the respiratory motion of the thorax and abdomen free of radiation. Respiratory motion information is vital for many medical applications in diagnostics [1], treatment planning [2] and execution [3]. Our application scenarios are MRI guided percutaneous interventions on the liver like radio frequency-, microwave- and cryoablation, biopsies, or brachytherapy, where the challenge of a moving target exists. 4D MRI methods have been proposed, but none satisfy all the needs for our interventional application. These needs are first, physiological correctness of the 4D sequence, and second, robustness against the out-of-plane motion. In this study, we propose a new 4D MRI reconstruction method. It utilizes retrospective sorting of dynamic 2D TRUFI MRI slices and is capable of imaging the whole liver during free breathing and capturing organ deformations caused by respiration. It reconstructs a physiologically meaningful sequence of respiratory states by utilizing a dedicated navigator frame and copes with out-of-plane motion.

## Related work

To our knowledge, there exist two approaches to acquiring 4D MRI, each with its unique advantages and disadvantages. The first is to acquire 3D MRI sequences in real-time, as done by Kim et al. [4] and Bled et al. [5]. The advantages of this approach are that it does not rely on gating and thus supports imaging events that do not occur repeatedly, i.e., events that are not periodic. The disadvantages of this approach are its low temporal and spatial resolution [6, 7] and its relatively small FOV, rendering it impossible to capture the respiratory motion of large organs like the liver.

The second approach is to reconstruct volumes for different organ states or breathing phases in retrospection by binning previously acquired data. Two main types of this approach exist. In the first type, the k-space data is sparsely sampled and binned before reconstructing a volume for a given organ state [8–10]. The strength of this type lies in capturing periodic organ state changes with a large FOV within a few minutes, depending on the length of the motion cycle. Its weaknesses are its assumption of strictly periodic organ motion. Thus, it can only reconstruct an average motion cycle of the target organ, which is not ensured to be physiologically meaningful. Furthermore, this type introduces image artifacts [11, 12] that could hinder motion estimation from the reconstructed 4D MRI.

The second type of the second approach reconstructs fast dynamic 2D sequences at all slice positions to cover the organ of interest. Then retrospective gating is applied to the resulting 2D images, binning them by different organ states, i.e., breathing states, and sorting them in their respective volumes. Its advantages are its applicability for non-periodic or quasi-periodic changes in the organ state and its high temporal and spatial resolution. Hence it is well-suited to capture motion variation, e.g., deep or shallow, abdominal or thoracic breaths within one session. It can work with a navigator or respiratory signal to ensure the physiological correctness of reconstructed motion. A further advantage of the binning strategy is its availability because it is readily usable with all MRI machines and all 2D sequences. Its disadvantages are that it is more time-intensive than the k-space binning and that much of the acquired data is redundant. The latter, however, can advantageously be used to increase the SNR of the reconstructed 4D images.

For both types, the surrogate can be intrinsic, relying on image information or k-space information, or extrinsic, relying on externally recorded signals, e.g., from using a breathing belt or form tracking markers that are placed on the abdomen of the subject. Siebenthal et al. [13, 14] utilize navigator slices as surrogate and vessel cross-section tracking as a matching

criterion. Cai et al. [15] use the body area. Lee et al. [16] use sagittal diaphragm profiles and reconstruct one breathing cycle. Tong et al. [17] propose a graph-based sorting where the weights are based on image information and semi-automatic assigned respiratory phase although, they are only able to reconstruct one best breathing cycle and not a variety of breathing cycles. Romaguera et al. [18] propose a graph-based approach using pseudo-navigators. A drawback of the graph-based navigator-less approach is that physiological correctness cannot be ensured even if temporal coherence is ensured.

## Materials and methods

We decided to follow the retrospective sorting approach because, as set out in the related work section, it is the only one suited for capturing physiologically meaningful, non-periodic organ motion with high temporal and spatial resolution and large field of views. Its only disadvantage is the long acquisition time, which can be overcome, as shown in this work. Specifically, we build upon the proposed method of von Siebenthal et al. [13, 14].

The Otto-von-Guericke-University Magdeburg ethics board approves our study "Studies with healthy subjects in 3 Tesla for methodological development of MRI experiments" (approval number 172/12), stating they concluded that there are no ethical concerns and that this approving assessment is made based on unchanged conditions. Oral and written consent was obtained during the study.

In the following three sections, we describe the general concept behind the baseline method and our method. In section Template updates and search region, we describe how we build upon the baseline to improve it and overcome the named drawbacks.

### MRI acquisition

Our MR data were acquired on a MAGNETOM Skyra MRI scanner (Siemens Medical Solutions, Erlangen, Germany). All images were acquired with a TRUFI sequence (TR = 39.96 ms, echo spacing = 3.33 ms, TE = 1.49 ms, flip angle = 30 degree, readout bandwidth = 676 Hz/px, base resolution = 176 $k_x$, phase resolution = 80% yielding a matrix size of 140 x 176, in-plane resolution 1.82mm x 1.82mm, out of plane resolution 4 mm, FOV: 255 mm x 320 mm). For faster measurement, a partial Fourier was used sampling 5/8 of the k-space asymmetrically in phase-encoding direction, i.e., roughly 60% of the $k_y$ lines, resulting in 88 actually acquired $k_y$ lines. Using this setup, we achieve acquisition times of 200 ms per slice. The acquisition setup was chosen to mimic an interventional setup as closely as possible. This specifically means high acquisition speed and just good enough contrast to detect the respiratory motion. No body array coil (surface array coil comprised of multiple elements) was used. Only the bore fixed receiver coil was used, which makes this 4D MRI method compatible with a wide range of external surrogates, including those that need a free line of sight to the abdomen of the subject. This includes, but is not limited to, surrogates based on a scan of the abdomen's surface or marker tracking on the abdomen. This is important to make the gathered motion information available for a wide range of interventional scenarios where different surrogates may be used to track breathing. A total of 19 data sets of 13 healthy subjects were acquired. One subject was imaged three times, four subjects were imaged twice, and eight subjects were imaged once. If a subject was imaged multiple times, then each data set acquisition was performed on different days to include variations that occur in between imaging sessions. Each data set consists of two reference sequences and several interleaved sequences. Both will be described in the following.

A reference sequence is a dynamic 2D MRI sequence of so-called navigator frames. A schematic depiction can be found in Fig 1. The navigator frames picture an image plane, in which

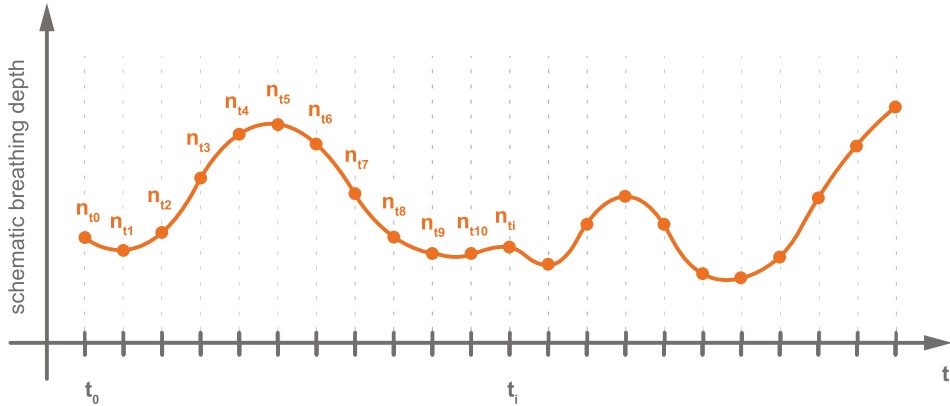

**Fig 1. Schematic depiction of a reference sequence.** A reference sequence shows a physiologically meaningful breathing curve and consists only of navigator frames that were imaged at the same slice position.

the respiratory motion is visible. In our case, we used a slice in the sagittal orientation that intersects the target organ—the liver—and shows vessel cross-sections, because their spatial distribution describes the breathing state well. This sequence is the reference for the 4D reconstruction. The reference contains a natural succession of different breathing patterns, like shallow or deep, thoracic or abdominal breathing, and is thus physiologically and profoundly meaningful. One reference sequence was acquired at the beginning and one at the end of each session. A reference sequence comprises 513 images (time points) covering a time of 102 seconds (about 20 breathing cycles).

Each interleaved sequence consists of equal parts of data frames and navigator frames (between 150 and 200 each), see Fig 2. The former are sorted into the 4D MRI sequences based on information extracted from the latter. Data slices and navigator slices were imaged alternatingly, facilitating the interleaved character of the sequence. The navigator slices are positioned exactly as in the reference sequence, rendering temporal reconstruction possible. The data slice sweeps over the target organ in 4 mm gaps during acquisition (see Fig 3), rendering spatial reconstruction possible. For each slice position of the reconstructed volume one

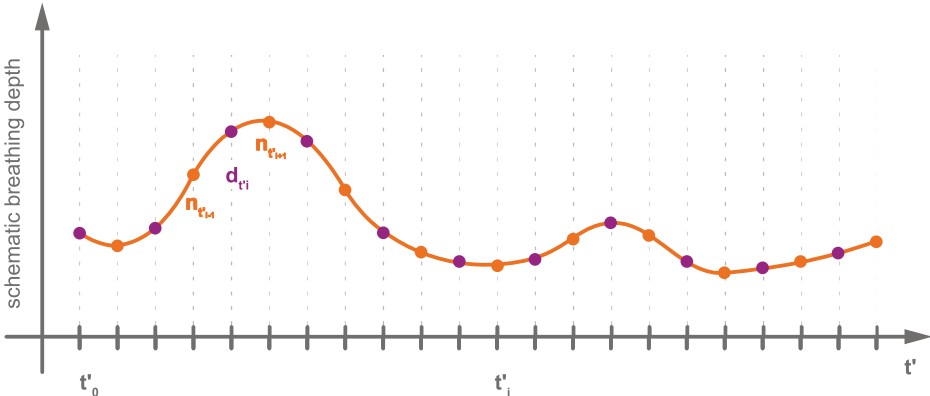

**Fig 2. Schematic depiction of an interleaved sequence.** An interleaved sequence consists of navigator frames and data frames that were imaged alternatingly. It shows a different breathing curve than the navigator sequence but contains similar breathing patterns.

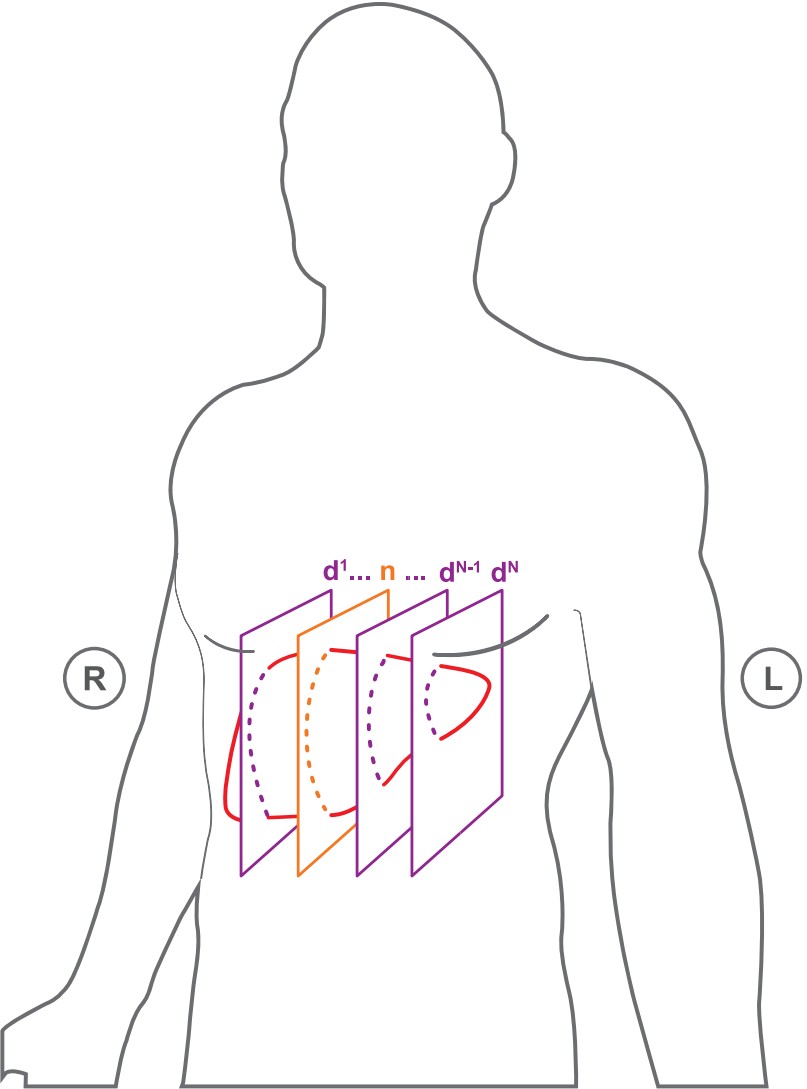

**Fig 3. Schematic depiction of slice positions capturing the target volume.** Slices are in sagittal orientation. The position of the navigator slice is the same for all sequences per subject. The slice positions for the data frames are distinct and correspond to different interleaved sequences from the 1'st to the N'th. Interleaved sequences are acquired from right to left.

interleaved sequences is acquired. The total number of interleaved sequences per subject ranges between 38 and 57 (mean = 46.68), depending on the size of the subjects' target organ to capture its whole volume. Thus, the total acquisition time for a subject ranged between 40 min and 80 min, excluding time for imaging localizers, determining navigator position and setting up the interleaved sequences. The total acquisition time is the time it took to capture all MRI images necessary for 4D MRI reconstruction, i.e., reference sequences and interleaved sequences. In the use case this acquisition would be made during planning before the actual intervention. The imaging of localizers, determining the navigator position and setting up the interleaved sequences took roughly 15 min per subject.

The acquisition time can be halved when using matching criteria that do not depend on a navigator slice. The total acquisition time can be further reduced by optimizing the acquisition

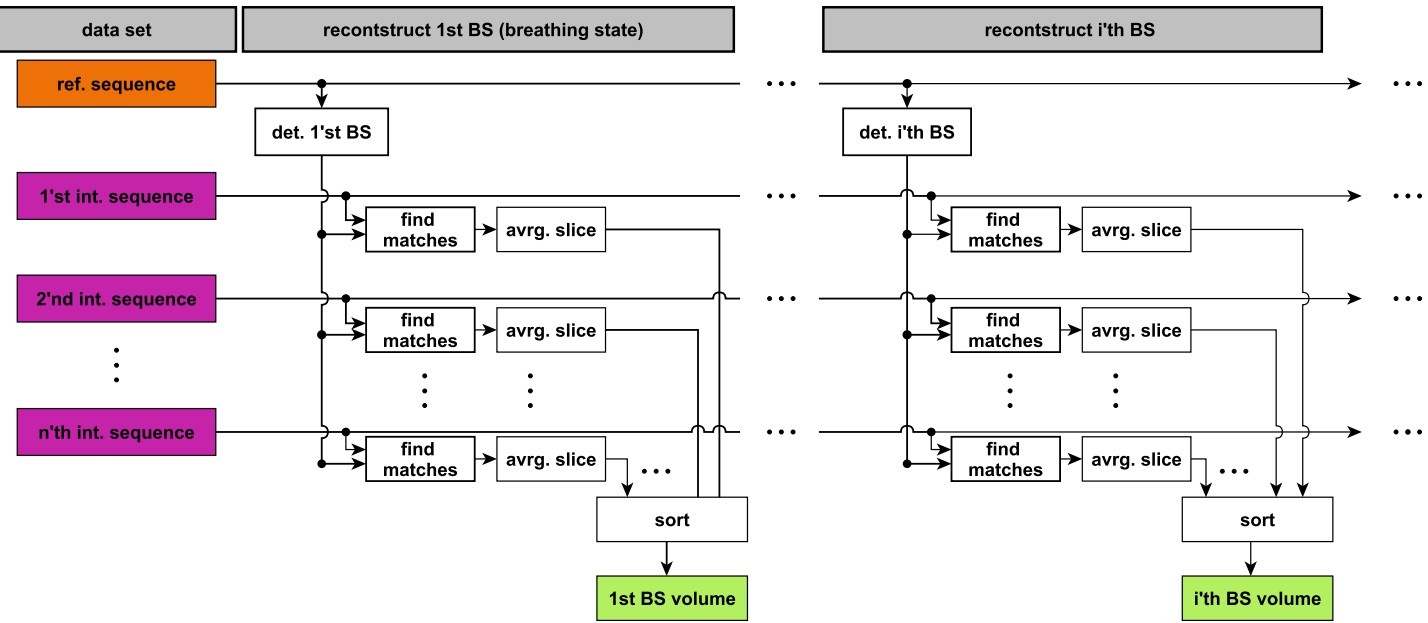

**Fig 4. Scheme of 4D MRI reconstruction.** For each time point in the reference sequence, a volume is reconstructed. For that in each interleaved sequence, the data slices are found that match the breathing state. They are then averaged and sorted into a volume.

scheme, allowing in-time breathing instructions for the subject for more efficient use of the acquisition time. During the intervention itself, only a surrogate, e.g., a navigator frame, has to be acquired in real-time as a query to the reconstructed 4D MRI or to a breathing model that was derived from the 4D MRI. All acquired MRI sequences used for 4D reconstruction, and a detailed acquisition protocol are publicly available [19].

## 4D MRI reconstruction

Our method and the baseline method use the reference sequence as grounds for the temporal reconstruction of a 4D MRI sequence showing a physiologically meaningful course of breathing states. The general scheme of the reconstruction process is depicted in Fig 4. For each time point in the reference sequence, i.e., for each frame, a volume is reconstructed. First, the breathing state of the frame is determined. Second, in each interleaved sequence, all data frames are found that match the breathing state, using a matching criterion, see Fig 5. Third, the found frames are averaged (binned) to one slice to improve the SNR (signal-to-noise ratio). Fourth, the averaged slice is inserted (sorted) into the volume at its designated position, which is known and unique for each interleaved sequence. Doing this for all reference frames results in a continuous 4D MRI sequence. The reconstructed FOV's range from 255 mm x 320 mm x 152 mm to 228 mm (140 x 176 x 38 to 57 voxels) depending on the size of the target organ. In the next section, the matching criterion is described in detail.

## Matching criterion

A matching criterion is used to find all data slices showing the reference breathing state within an interleaved sequence. The respiratory state of a frame is determined by its enclosing navigator frames. Hence, the matching criterion acts on pairs of navigator frames that encase another frame (navigator or data frame); see brackets in Fig 5. It is based on the displacement of

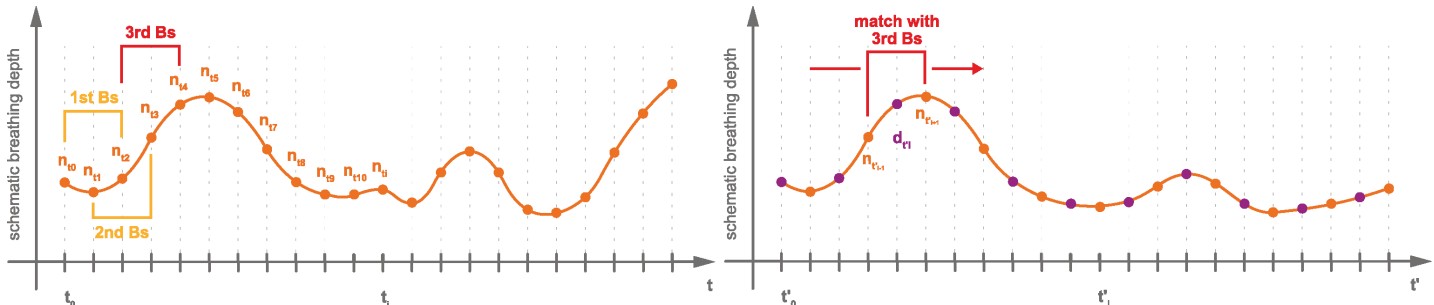

**Fig 5. Scheme of finding data slices that match specific breathing state.** On the left hand, the reference sequence is depicted. The red bracket represents the third breathing state. It is found in the interleaved sequence, depicted on the right, by comparing the enclosing navigator slices.

tracked vessels within the navigator frames. Assume a navigator frame $n_{t_i}$ at time point $t_i$ in the reference sequence that shows a reference breathing state $BS_r$. We want to find a data frame $d_{t_j}$ with the same breathing state as $n_{t_i}$. To this end, the enclosing navigator frames of both $d_{t_j}$ and $n_{t_i}$ are used. The enclosing navigator frames of $d_{t_j}$ are $n_{t_{j-1}}$ and $n_{t_{j+1}}$ and the enclosing frames of $n_{t_i}$ are $n_{t_{i-1}}$ and $n_{t_{i+1}}$. The vessel displacements from $n_{t_{j-1}}$ to $n_{t_{i-1}}$ and from $n_{t_{j+1}}$ to $n_{t_{i+1}}$ are calculated. When the sum of all vessel displacements for two pairs of navigator frames is under a certain threshold, then the two enclosed frames are assumed to be a match, i.e., to show the same breathing state. The threshold is the only parameter of the method. It determines the maximally allowed displacements for two frames to be counted as a match.

The vessel tracking is realized via template matching using OpenCV [20] and its similarity measure TM_CCOEFF_NORMED (see Eq 1).

$$R(x, y) = \frac{\sum_{x', y'} (T'(x', y') \cdot I'(x + x', y + y'))}{\sqrt{\sum_{x', y'} T'(x', y')^2 \cdot \sum_{x', y'} I'(x + x', y + y')^2}} \tag{1}$$

where

$$T'(x', y') = T(x', y') - 1/(w \cdot h) \cdot \sum_{x'', y''} T(x'', y'')$$
$$I'(x + x', y + y') = I(x + x', y + y') - 1/(w \cdot h) \cdot \sum_{x'', y''} I(x + x'', y + y'') \tag{2}$$

Here $T'$ is the template $T$ minus its mean pixel intensity, and $I'$ is an image patch with the same size as the template. Its pixel values are also shifted by minus the patches mean pixel intensity. $w$ and $h$ are the width and height of the template and the patch.

$R$ is the resulting image of the template matching. Each entry $R(x, y)$ contains the similarity value of the template to the source image at position $(x, y)$.

The templates are manually defined for each tracked vessel cross-section in the reference sequence. To this end, a user identifies trackable vessels in one slice of the reference sequence prior to the 4D reconstruction, which takes only a few seconds. In our case, trackable means that the vessel cross-section or cluster of cross-sections will be visible in most navigator frames throughout the whole navigator sequence and that the cross-section has a high contrast to the surrounding tissue as well as a high signal to noise ratio. This is mostly not the case for small cross-sections but true for larger ones.

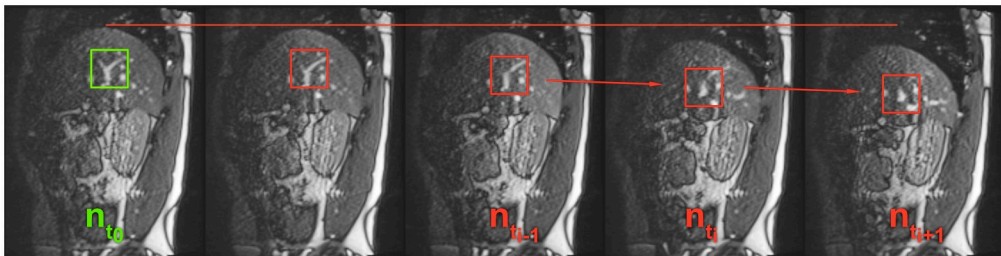

**Fig 6. Out-of-plane motion and template updates.** The figure shows a series of navigator slices. The green rectangle denotes a typical ROI that was manually determined as a template. In the red rectangles, one can see how the vessel cross-section changes its appearance during the breathing cycle. For viewing purposes only, the images gradation curve was altered globally to enhance contrast.

## Template updates and search region

One of the challenges for the template matching is the out-of-plane motion of the vessel cross-sections in the navigator frames. In these cases, the searched-for regions are changing their appearance throughout breathing; hence, they are difficult to find using fixed templates.

To increase robustness against the out-of-plane motion, we propose to apply template updates within the reference sequence. In Fig 6, one can see how the appearance of a vessel cross-section can change during a breathing cycle. The method starts with the templates that were defined manually on reference frame $n_{t_0}$. Then, for each following navigator frame $n_{t_i}$ that was captured at time point $t_i$, the templates get automatically updated, as follows: The positions of all tracked vessels in $n_{t_i}$ are found with subpixel precision using the templates from time point $t_{i-1}$. Then a new set of templates is cut from $n_{t_i}$ based on the position of the matched templates. The template position is updated with floating-point precision. The updates ensure that changes in the appearance of the tracked vessel are represented in the updated templates. The subpixel precision in the updates is needed to avoid drift during the update.

Another concern of the reconstruction approach is speed. In its original form, the method matches each template against each navigator frame, resulting in a substantial computational burden. We propose to speed up the vessel tracking by exploiting spatial coherence between temporally adjacent navigator frames. The underlying assumption is that the next searched-for match is in a small spatial neighborhood around the previously found match, which is the case due to fast and continuous acquisition. Therefore, we only use a small neighborhood around the last matched template position as a search area.

Moreover, we automatically detect breathing states that cannot be reconstructed entirely and use that knowledge to inform where (temporally and spatially) the 4D sequence is incomplete. This information is essential for the later application, because of the visual feedback that can be provided to the physician in real-time when the motion information is insufficient to fuse the planning data to the interventional data.

## Evaluation

We compare our method with the baseline method of Siebenthal et al. through reconstruction rate and image quality. We define the reconstruction rate as the percentage of the number of slices in the volume that could be reconstructed by the method. Note that this does not account for false positives or false negatives because the ground truth is not available to us. We also investigate how the acquisition order of the reference sequence and interleaved sequence

**Table 1. Tested parameter values.**

| Parameter | Value |
|---|---|
| Threshold | 0.5; 1; 2 |
| Similarity measure | TM_CCORR_NORMED; TM_CCOEFF_NORMED |
| Reference Sequence | ref. 1; ref. 2 |

influences the method's ability to find matching data frames. We evaluate the point of false positives indirectly using a qualitative assessment of both approaches. The image quality is assessed in a double-blind study with interventional radiologists.

**Reconstruction rate.** We compare the reconstruction rate of both methods for different parameterizations. This is possible because the baseline method uses the same parameters in its matching criterion. When a subject was imaged multiple times, the reconstruction rates of its respective data sets were averaged for the statistical analysis to avoid possible biases. We tested the parameters shown in Table 1. We tested the threshold, for the values 0.5, 1, and 2. Evaluating different thresholds from a quantitative point-of-view allows us to judge which method will be more suitable for different applications that differ in the kind of trade-off between precision and coverage that is preferable in the application. With lower (stricter) thresholds, the coverage goes down and the precision increases. With higher thresholds, the coverage increases and the precision decreases. We tested two similarity measures from OpenCV, namely TM_CCOEFF_NORMED (see Eq 1) and TM_CCORR_NORMED (see Eq 3), and we tested the influence of the chosen reference sequence, ref. 1 and ref. 2, where ref. 1 is acquired before and ref. 2 is acquired after the interleaved sequences.

$$R(x, y) = \frac{\sum_{x',y'}(T(x', y') \cdot I(x + x', y + y'))}{\sqrt{\sum_{x',y'}T(x', y')^2 \cdot \sum_{x',y'}I(x + x', y + y')^2}} \tag{3}$$

where T is the template, I is the image and R is the resulting image with the highest intensity in position $(x, y)$, where the similarity was the highest.

A four-factorial analysis of variance (ANOVA) was conducted to test for the effects of the aforementioned factors on the reconstruction rate.

**Reconstruction quality.** We conducted a double-blind study with ten interventional radiologists to compare the reconstruction quality of both methods and to evaluate whether our method's reconstruction quality improves over the baseline. Participants were recruited from a General Radiology clinic. Their professional experience ranged from 4 months to 20 years (median: 16 months, mean: 62 months).

The interviews were in no way invasive, and no data that would allow for participant identification was included in the analysis. Thus, IRB approval was not requested for the interviews. In all cases oral participation consent was obtained and recorded.

Each radiologist was shown a set of 48 slice image pairs. The images of a pair were reconstructed from the same subject and breathing state showing the same anatomical structure and having the same slice position and orientation. One slice in a pair was sampled from a reconstruction of the baseline method. The other was sampled from a reconstruction of our method. Slices of a reconstructed volume are depicted in Fig 7. The radiologists had to decide which of the images in a pair shows the anatomy of the target organ more faithfully, i.e., with fewer image artifacts. Participants did not see the two slices from each pair simultaneously but could switch back and forth between them as often as they wanted before picking one. Participants were asked to select the slice they considered better. A neutral option was provided. For the

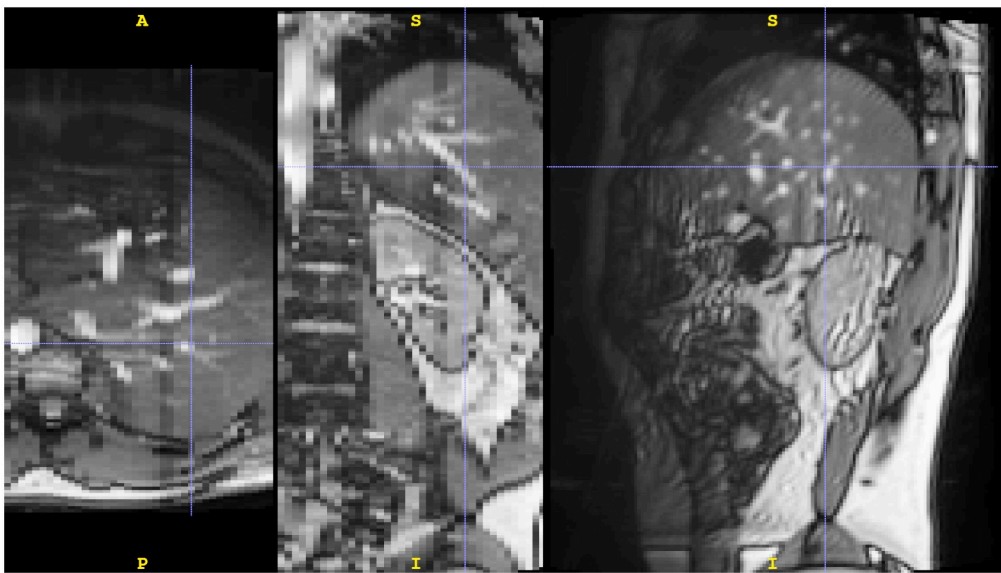

**Fig 7. Axial, coronal and sagittal slices of a reconstructed volume.** The images gradation curve was altered globally to enhance contrast for better viewing only. In the axial and coronal orientation, one can see that our method is capable of reconstructing smooth and continuous volumes from sagittal slices.

evaluation of reconstruction quality, the parameter set was chosen to be 1 px threshold and TM_CCOEFF_NORMED as a similarity measure for both methods. Only those volumes were considered for comparison, for which both methods had a reconstruction rate of at least 80%. For each radiologist, 48 volume pairs were chosen randomly.

Furthermore, in both volumes, we automatically masked slices out (setting intensity values to black), where either of the methods did not find a matching data frame. We made both volumes identical in the amount and distribution of black slices. This was done because it is likely that a reduced reconstruction rate for a volume would be detrimental to its perceived reconstruction quality. Each slice pair was sampled at a random orientation and position chosen within a range, such that the sampled slice would show the target organ. Slices were sampled either in sagittal, coronal, or axial orientation. Due to a software error, the number of slices for different planes was slightly imbalanced: Overall, 100 slices were shown for the sagittal and axial orientation each, and 280 slices were shown for the coronal orientation. For each of the 480 image pairs shown to participants, we recorded which method was preferred, if either. For data analysis, the two methods were appointed one 'point' each for every time they had been preferred. For each neutral vote, both methods were appointed a half 'point'. This led to a dichotomous variable that allows for a direct comparison of the two methods' scores. A one-sided binomial test was conducted ($H_0$: $p_{our\_method} \leq 0.5$, $H_1$: $p_{our\_method} > 0.5$).

## Results

Table 2 shows the mean reconstruction rates for all parameter combinations. Our method has a consistently higher reconstruction rate than the baseline (about twice as high) for all parameter sets. Figs 8 and 9 show the respective distribution of reconstruction rates. All underlying reconstruction rates per reconstructed 4D MRI and all tested parameters are provided in S1 File.

**Table 2. Mean reconstruction rates [%] of our method and baseline.** Reconstruction rates are given in percent reconstructed of a volume. Bold is the best rates for each parameter set.

| threshold | | TM_CCORR_NORMED | | | TM_CCOEFF_NORMED | | |
|---|---|---|---|---|---|---|---|
| | | 2px | 1px | 0.5px | 2px | 1px | 0.5px |
| ref. 1 | baseline | 24.58 | 15.95 | 9.94 | 41.78 | 24.10 | 12.74 |
| | our method | **73.60** | **40.99** | **23.24** | **77.69** | **47.10** | **27** |
| ref. 2 | baseline | 46.86 | 31.95 | 18.75 | 60.09 | 40.07 | 22.92 |
| | our method | **79.67** | **56.89** | **36.78** | **82.18** | **58.53** | **37.34** |
| avrg. | baseline | 35.72 | 23.95 | 14.34 | 50.93 | 32.08 | 17.83 |
| | our method | **76.63** | **48.94** | **30.01** | **79.93** | **52.82** | **32.17** |

The four-factorial ANOVA showed significant main effects for all four factors and one significant interaction effect for the reconstruction method and the threshold used (Table 3). This interaction effect describes that while our method performs better than the baseline method at all threshold levels, it achieves more significant improvements at higher thresholds (see also Figs 8 and 9).

On the tested data, it was also more robust against the chosen similarity measure used for the template matching and also more robust against whether the reference sequence was acquired in the beginning or at the end of the session. Though, these interaction effects could not be shown to be significant in the ANOVA.

A correlation between acquisition order of the slice positions relative to the reference sequence and the ability of the methods to reconstruct these slice positions can be seen in Fig 10. With the increasing temporal distance between the acquisition of an interleaved sequence and the reference sequence, both methods find fewer similar slices for the corresponding slice position. Reference sequence one (red graphs) is acquired before the interleaved sequences. Here both methods find more slices for the earlier slice positions. Reference sequence two (blue graphs) is acquired after all interleaved sequences. Here both methods find more slices for the later slice positions.

**Fig 8. Reconstruction rates for reference sequence one.**

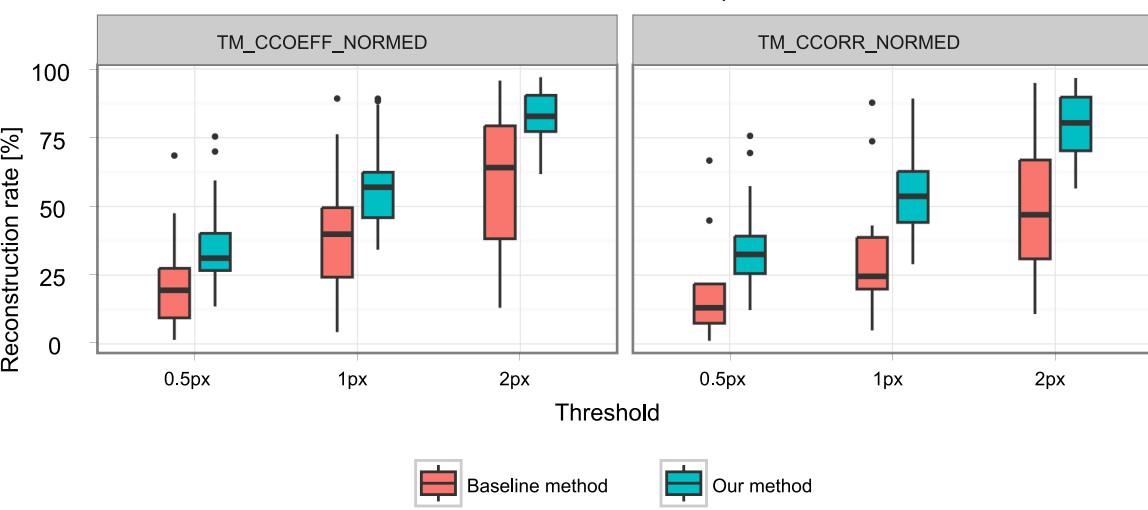

**Fig 9. Reconstruction rates for reference sequence two.**

The mean reconstruction time of our method is 24.19 seconds, with a standard deviation of 6.82 seconds. The mean reconstruction time of the baseline is 73 seconds, with a standard deviation of 21.81 seconds.

In the double-blind study, overall, participants selected our method in 156 trials, the baseline method in 111 trials, and had no preference in 213 trials (see Fig 11). Following our analysis method, this yielded 262.5 'points' for our method and 217.5 'points' for the baseline method (p = 0.02). All acquired data of the study is provided in S2 File.

The study shows that radiologists perceive the reconstruction quality of our method as significantly better than the baseline method, although the effect seems to be small.

## Discussion and conclusion

The particular acquisition scheme shows difficulties with changes in breathing patterns that arise over a more extended period, like the typical flattening of the resting breath. Slice positions to the left are imaged only at the end of acquisition time, whereas slices to the right are only imaged at the beginning. As a consequence, if the reference sequence was captured in the beginning, it can show breathing states that do not occur later, when slice positions to the left are imaged. Deep breaths often can not be fully reconstructed since image data of the left slice

**Table 3. Main results of the ANOVA on the reconstruction rate.**

| Effect type | Factor | df | F | p |
|---|---|---|---|---|
| Main effects | *Reconstruction method* | 1 | 134.99 | **<0.001** |
| | *Threshold* | 2 | 106.56 | **<0.001** |
| | *Similarity measure* | 1 | 8.33 | **0.004** |
| | *Reference sequence* | 1 | 37.40 | **<0.001** |
| Interaction effect | *Reconstruction method * Threshold* | 2 | 7.71 | **<0.001** |
| | *Rec method * Similarity measure* | 1 | 1.95 | 0.164 |
| | *Rec method * Reference sequence* | 1 | 1.41 | 0.236 |

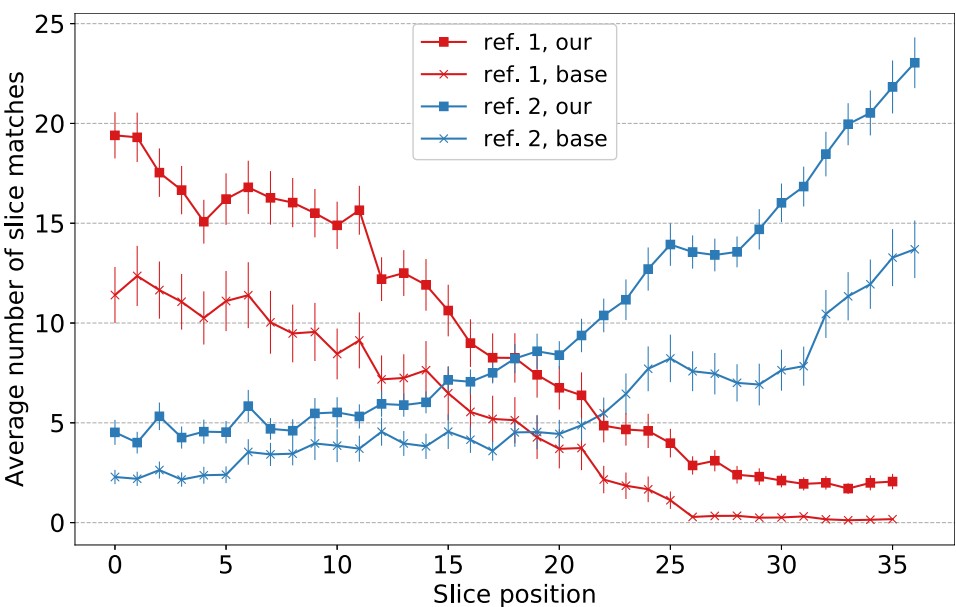

**Fig 10. Correlation of slice position and number of slice matches.** Red graphs represent the average number of slice matches for the first reference sequence (averaged over all subjects). Blue graphs correspond likewise to the second reference sequence. Graphs with squares represent our method; graphs with crosses represent the baseline method. Error bars represent standard deviation and are scaled by 0.1 for better readability.

positions was not acquired for deep breathing states. Generally speaking, the scheme has difficulties with breathing states that are less frequent. This problem can be solved in changing the acquisition scheme. Instead of first acquiring all slices in one position before moving on to the next slice position, it is beneficial to move the slice position after each acquisition while keeping the navigator position fixed. This rotating acquisition scheme could also be combined with intermediate reference sequences. This would directly counter the problem with flattening

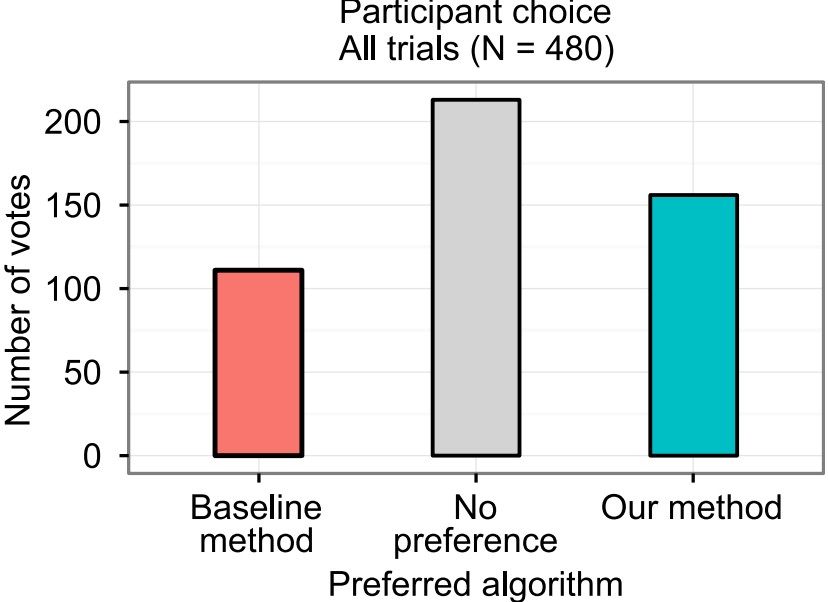

**Fig 11. Participant choice.** The bars represent the number of times each option was chosen out of 480 trials.

breath over time. Furthermore, with the new scheme, it is feasible to give a few commands, so the subject can take a few more deep breaths in the beginning before starting to relax more.

The rotating acquisition scheme was used by Siebenthal et al. on a 1.5T Philips Intera whole-body MRI system [14]. However, Siemens MRI machines do not allow this kind of scheme. A solution to the problem that is independent of the scanner used is to use external respiratory signals instead of navigator frames. Preiswerk et al. [21] had correlated 1D MR compatible ultrasound with 2D and multiplanar MRI. This allows for the continuous rotating acquisition of the data slices on any MRI machine. Celicanin et al. [22] propose a simultaneous multislice (SMS) imaging technique that allows for the simultaneous acquisition of navigator and data frames, increasing the temporal coherence of navigator and data frame. Barth et al. [23] give a current overview of parallel imaging and SMS imaging techniques. These would integrate well with the rotational acquisition scheme when using body array coils. No body array coil is used in our experiment to ensure a line of sight for external marker tracking. However, when external marker tracking is not needed, a body array coil can readily be used in conjunction with our method to have better image contrast and possible faster imaging with aforementioned SMS techniques applied. When flat, flexible array coils with an opening for operation become available, those benefits, i.e, higher SNR, faster acquisition and line of sight, could be combined.

Regarding the acquisition time, the aforementioned changes to the acquisition scheme would half the acquisition time in our case to between 20 and 40 min.

Regarding the reconstruction rate, because of the lack of ground truth, it is not possible to account for false negatives and false positives in the evaluation. Based on this fact, the reconstruction rate of both methods will possibly be higher than measured in this study. This is because, in our test data, the number of reconstructable slice positions in each volume is lower than the number of slices in a volume, resulting from the acquisition scheme mentioned above.

An open issue arises when vessel cross-sections in the navigator frame are not continually visible. This frequently happens to depend on blood flow. To solve this, one could detect outliers in the template matching step and omit those for the calculation of the summed displacement.

We decided to use MRI data of healthy volunteers for the development and evaluation of the method. For a proof of concept of our method, this eliminates possible adverse effects of liver diseases on the respiration of the patient, making the evaluation environment more controlled. However, in future work, it has to be evaluated if typical diseases targeted by this method, like liver carcinoma, affect the method. This could be especially the case if the disease impairs the respiration of the patient. If the patient's breathing shows no or few repetitions of patterns, this would be a challenge for the method because whilst allowing for irregular breathing, it assumes that patterns are recurring over time.

In its presented form, our method relies on a manual step in which the ROIs around the vessel cross-sections are defined. In a real clinical setting, this is intended to be done offline after the planning MRI session and before the date of the intervention on a suitable computer, not directly on the MRI machine. Even though this manual interaction is minimal and takes less than a minute to perform, it could and should be automated in future work. This could be solved as a classification problem in image space using the temporal information of the reference sequence as supporting information.

In our evaluation of the visual reconstruction quality, we only compare our method relative to the baseline. The provided neutral option does not differentiate between equally good and equally bad or unusable, and no absolute data was gathered. Hence, our analysis does not

show whether the reconstructions are good enough for a given task or not. The analysis only indicates that our method's reconstruction quality improves over the baseline.

In summary, our results clearly show that template updates are an effective and efficient means to increase reconstruction rates and image quality of the reconstruction result for template-based 4D MRI reconstruction methods. We reported that employing search regions significantly reduces reconstruction time. The results suggest that our method is preferable compared to the baseline. This is regardless of the application's favorable trade-off between precision and coverage because, in all cases, reconstruction rates are higher than the baseline.

## Supporting information

**S1 File. Reconstruction rates.** Reconstruction results of the experiments for all 4D MRI reconstructions and tested parameters.
(CSV)

**S2 File. Study results.** Participants choices in the image quality study.
(CSV)

**S1 Data.**
(PDF)

## Acknowledgments

We gratefully acknowledge the Research Campus STIMULATE Solution Centre for Image Guided Local Therapies for providing the MRI scanner for our study and Cindy Lübeck for operating the scanner.

## Author Contributions

**Conceptualization:** Gino Gulamhussene, Marko Rak, Christian Hansen.

**Data curation:** Gino Gulamhussene, Fabian Joeres, Maciej Pech.

**Formal analysis:** Gino Gulamhussene, Fabian Joeres, Marko Rak, Maciej Pech.

**Funding acquisition:** Christian Hansen.

**Investigation:** Gino Gulamhussene, Fabian Joeres.

**Methodology:** Gino Gulamhussene, Fabian Joeres, Marko Rak, Christian Hansen.

**Project administration:** Gino Gulamhussene, Marko Rak.

**Resources:** Gino Gulamhussene, Maciej Pech, Christian Hansen.

**Software:** Gino Gulamhussene, Marko Rak.

**Supervision:** Marko Rak, Maciej Pech, Christian Hansen.

**Validation:** Gino Gulamhussene, Fabian Joeres, Marko Rak.

**Visualization:** Gino Gulamhussene, Fabian Joeres.

**Writing – original draft:** Gino Gulamhussene, Fabian Joeres, Marko Rak.

**Writing – review & editing:** Gino Gulamhussene, Fabian Joeres, Marko Rak, Maciej Pech, Christian Hansen.

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
