## [Decision Letter · Decision Letter 0]

11 Sep 2019

PONE-D-19-18726

4D MRI: Robust sorting of free breathing MRI slices for use in interventional settings

PLOS ONE

Dear Mr. Gulamhussene,

Thank you for submitting your manuscript to PLOS ONE. After careful consideration, we feel that it has merit but does not fully meet PLOS ONE’s publication criteria as it currently stands. Therefore, we invite you to submit a revised version of the manuscript that addresses the points raised during the review process.

Additionally, please be aware of PLOS ONE policy for making the published research as reproducible as possible, in particular for what it concerns the availability of data. In order for your research to be published please elaborate on the reasons you are not allowed to share data (even anonymized) and/or the criteria upon which researchers can be granted access.

We would appreciate receiving your revised manuscript by Oct 26 2019 11:59PM. To enhance the reproducibility of your results, we recommend that if applicable you deposit your laboratory protocols in protocols.io, where a protocol can be assigned its own identifier (DOI) such that it can be cited independently in the future. For instructions see: http://journals.plos.org/plosone/s/submission-guidelines#loc-laboratory-protocols

We look forward to receiving your revised manuscript.

Kind regards,

Enrico Grisan

Academic Editor

PLOS ONE

Journal Requirements:

'This study was supported by funding from the Investitionsbank Sachsen-Anhalt (https://www.ib-sachsen-anhalt.de/) to GG with the grant number 1704/00038. The funder had no role in the study design, data collection and analysis, decision to publish, or preparation of this manuscript.'

We note that you received funding from a commercial source: Investitionsbank Sachsen-Anhalt

Additional Editor Comments (if provided):

Dear Authors,

the reviewers' have found scientific merit on your work and we are requesting a minor revision to address the point raised in their comments.

However, PLOS-ONE request the data to be made available, and it is not clear how anonymized MRI data can pose ethical challenges (just thinking about the many repositories targeting variuos diseases). In order to allow the publications, you should provide strong reasons why MRI data can not be shared and clarify the criteria for access.

Reviewers' comments:

Reviewer's Responses to Questions

**Comments to the Author**

1. Is the manuscript technically sound, and do the data support the conclusions?

Reviewer #1: Yes

Reviewer #2: Yes

2. Has the statistical analysis been performed appropriately and rigorously? 

Reviewer #1: Yes

Reviewer #2: Yes

3. Have the authors made all data underlying the findings in their manuscript fully available?

Reviewer #1: No

Reviewer #2: No

4. Is the manuscript presented in an intelligible fashion and written in standard English?

Reviewer #1: Yes

Reviewer #2: Yes

5. Review Comments to the Author

Reviewer #1: The paper presents a method for the reconstruction of 4D MRI sequences accounting for many states of respiratory motion. It finds that the proposed method, compared to a baseline, reconstructs sequences more often, more quickly, and to a higher perceived quality. Overall, while the method is a fairly incremental improvement over existing work, the paper is clearly written and the experiments well conceived.

I think there are a few minor issues with the paper which should be corrected in revisions.

- The two similarity measures used are referenced by their names in OpenCV - TM_CCORR_NORMED and TM_CCOEFF_NORMED.These measures should be described (mathematically) in the text. They are a key part of the method and discussed at length, but a reader who is unfamiliar with OpenCV software would have no idea what they are. Their exact implementations in OpenCV may even change in the future.

- There are a number of grammatical / spelling errors in the text which should be corrected. For example,

-- the sentence beginning on line 5 should read "Our application scenario[s] are ... where the challenge of a moving target exist[s] ... but none satisfy all [the] needs ..." (corrections in [])

-- the sentence beginning on line 108 seems to have had some words cut out of it

-- the caption for Figure 6 misspells 'contrast'

- Regarding the data availability, it is reasonable to withhold the actual MRI image data - the journal suggests though that the numerical scores used to generate the summary statistics in, for example, tables 2 and 3 could be provided.

Reviewer #2: The authors describe a method for reconstructing four-dimensional volumes (three spatial + one temporal) from multiple 2D acquisitions. Navigator images are interleaved with imaging frames to provide a method for assigning 2D slices to the 4D volume. The work is built upon a method from Siebenthal et al and describes a minor improvement by constraining the search space for the navigator tracking. The data are not publicly available but the authors do note it is available upon reasonable request.

Please find minor comments below:

1). Please contextualize the imaging time (40 to 80 minutes) for interventional imaging. For non-interventionists, this seems like a very long acquisition time.

2). Please comment on why only healthy volunteers were scanned and speculate on what limitations the method may face in the presence of irregular breathing that is more likely in a patient population.

3). Please briefly describe TM CCOEFF NORMED and TM CCORR NORMED

4). Please comment on the limitations of manual steps such as ROI selection in the scanner environment. Could such steps be automated?

5). Please provide reconstruction times in the body of the paper.

6. PLOS authors have the option to publish the peer review history of their article (what does this mean?). If published, this will include your full peer review and any attached files.

Reviewer #1: No

Reviewer #2: No

---

## [Author Response · Author response to Decision Letter 0]

30 Oct 2019

We thank PLOS ONE’s academic Editor Enrico Grisan and the anonymous reviewers for their 

comments. We have attempted to address all remarks in the revised version of the manuscript. 

Below we give detailed responses to all reviewers comments and describe the corresponding 

changes in our manuscript. 

Enrico Grisan Academic Editor: 

01. Please be aware of PLOS ONE policy for making the published research as reproducible as 

possible, in particular for what it concerns the availability of data. In order for your research 

to be published, please elaborate on the reasons you are not allowed to share data (even 

anonymized) and/or the criteria upon which researchers can be granted access.:

We submit the anonymized data set underlying the computed scores as supporting 

information in a csv file with the revised submission. We are furthermore providing 

the anonymized MRI image data through the open repository of the university library 

of the Otto-von-Guericke-University Magdeburg under the creative commons license 

CC-BY-SA (DOI: 10.24352/UB.OVGU-2019-093; https://doi.org/10.24352/UB.OVGU-2019-

093) 

Journal Requirements: 

02. Please ensure that your manuscript meets PLOS ONE's style requirements, including those 

for file naming.: 

We made sure to follow PLOS ONE's style requirements. In particular, we made sure 

that figure references follow the style guides when referring to more than one figure 

at once and made changes to the manuscript accordingly. We made changes so that 

figures are cited in order of appearance. We checked that all images adhere to the file 

naming rules of PLOS ONE. 

03. We note that you have indicated that data from this study are available upon request. 

PLOS only allows data to be available upon request if there are legal or ethical restrictions 

on sharing data publicly.: 

Please see our answer to point 01. 

04. We note that you received funding from a commercial source: Investitionsbank Sachsen-

Anhalt: 

We believe there is a misunderstanding. The Investitionsbank Sachsen-Anhalt is an 

institute of the State of Saxony-Anhalt that has only administrative functions to 

distribute public funding. It is not a commercial funder. 

Reviewer 1 

05. The two similarity measures used are referenced by their names in OpenCV - 

TM_CCORR_NORMED and TM_CCOEFF_NORMED.These measures should be described 

(mathematically) in the text. They are a key part of the method and discussed at length, 

but a reader who is unfamiliar with OpenCV software would have no idea what they are. 

Their exact implementations in OpenCV may even change in the future.: 

Thank you for pointing that out. We added mathematical formulas and explanations 

in the manuscript (cf. page 5 lines 147-153 and page 7 lines 210-212) to make the 

used similarity measures more transparent to readers. 

06. There are a number of grammatical / spelling errors in the text which should be corrected. 

For example: 

a. the sentence beginning on line 5 should read "Our application scenario[s] are ... 

where the challenge of a moving target exist[s] ... but none satisfy all [the] needs 

..." (corrections in []): 

Thank you for pointing that out. We corrected the sentence. 

b. the sentence beginning on line 108 seems to have had some words cut out of it 

Thank you for pointing that out.: We corrected the sentence. 

c. the caption for Figure 6 misspells 'contrast' 

Thank you for pointing that out.: We corrected the sentence. 

 We also have had the revised manuscript proofread by a native speaker. 

Regarding the data availability, it is reasonable to withhold the actual MRI image data - the 

journal suggests though that the numerical scores used to generate the summary statistics 

in, for example, tables 2 and 3 could be provided.: 

Thank you for that point. Please see our answer to point 01. 

Reviewer 2 

07. Please contextualize the imaging time (40 to 80 minutes) for interventional imaging. For 

non-interventionists, this seems like a very long acquisition time.: 

Thank you for pointing that out. We made changes to the manuscript (cf. page 4 lines 

105 - 118) to better point out how the total acquisition time is to understand. See 

lines 107 to 122 in the revised manuscript with marked-up changes. 

08. Please comment on why only healthy volunteers were scanned and speculate on what 

limitations the method may face in the presence of irregular breathing that is more likely in 

a patient population.: 

Thank you for pointing that out. We changed the manuscript (cf. page 10 lines 322 - 

330) to address this question. See lines 335 to 343 in the revised manuscript with 

marked-up changes. 

 09. Please briefly describe TM CCOEFF NORMED and TM CCORR NORMED: 

Thank you for pointing that out. Please see our answer to point 05. 

10. Please comment on the limitations of manual steps such as ROI selection in the scanner 

environment. Could such steps be automated?: 

Thank you for pointing that out. We changed the manuscript (cf. page 10 lines 331 – 

338) to address the limitations of manual steps in our method. See lines 344 to 351 in 

the revised manuscript with marked-up changes. 

11. Please provide reconstruction times in the body of the paper.: 

Thank you for pointing that out. We changed the manuscript (cf. page 9 lines 272 - 

274) to provide mean reconstruction times of both methods as well as the respective 

standard deviations. See lines 284 to 286 in the revised manuscript with marked-up 

changes.

---

## [Decision Letter · Decision Letter 1]

6 Apr 2020

PONE-D-19-18726R1

4D MRI: Robust sorting of free breathing MRI slices for use in interventional settings

PLOS ONE

Dear Mr. Gulamhussene,

Thank you for submitting your manuscript to PLOS ONE. After careful consideration, we feel that it has merit but does not fully meet PLOS ONE’s publication criteria as it currently stands. Therefore, we invite you to submit a revised version of the manuscript that addresses the points raised during the review process.

Both reviewers are enthusiastic for the manuscript. However, there are few minor comments to be addressed. Please respond all carefully. 

We would appreciate receiving your revised manuscript by May 21 2020 11:59PM. To enhance the reproducibility of your results, we recommend that if applicable you deposit your laboratory protocols in protocols.io, where a protocol can be assigned its own identifier (DOI) such that it can be cited independently in the future. For instructions see: http://journals.plos.org/plosone/s/submission-guidelines#loc-laboratory-protocols

We look forward to receiving your revised manuscript.

Kind regards,

Haydar Celik, PhD

Academic Editor

PLOS ONE

Reviewers' comments:

Reviewer's Responses to Questions

**Comments to the Author**

1. If the authors have adequately addressed your comments raised in a previous round of review and you feel that this manuscript is now acceptable for publication, you may indicate that here to bypass the “Comments to the Author” section, enter your conflict of interest statement in the “Confidential to Editor” section, and submit your "Accept" recommendation.

Reviewer #2: All comments have been addressed

Reviewer #3: All comments have been addressed

2. Is the manuscript technically sound, and do the data support the conclusions?

Reviewer #2: Yes

Reviewer #3: Partly

3. Has the statistical analysis been performed appropriately and rigorously? 

Reviewer #2: Yes

Reviewer #3: No

4. Have the authors made all data underlying the findings in their manuscript fully available?

Reviewer #2: Yes

Reviewer #3: Yes

5. Is the manuscript presented in an intelligible fashion and written in standard English?

Reviewer #2: Yes

Reviewer #3: Yes

6. Review Comments to the Author

Reviewer #2: (No Response)

Reviewer #3: The manuscript titled “4D MRI: Robust sorting of free breathing MRI slices for use in interventional settings” presents a modification of a previously published technique for respiratory motion extraction and 4D image volume generation as pre-procedural imaging for MR-guided interventions. The manuscript is well written, and the results presented in a clear manner. Nevertheless, the manuscript could benefit from direct and succinct clarification as to what are the differences between “their method” and the “reference method” since their method builds upon the reference one. The Methods section clarifies the method, but it is not clear where the original ends and where the new one begins. This should be crystal clear for any reader without having to read the entirety of the work.

General Comments

1) Treatments of individual subjects: The authors acquire multiple datasets from some subjects (“One subject was imaged three times, four subjects were imaged twice…”). These repeated acquisitions are treated as independent measurements and it appears there is no distinction when calculating metrics and averages or in any of the statistical analysis. Yet, these samples are not statistically independent measures. It is clear that in MRI some volunteers naturally provide excellent images and others are simply less photogenic (usually move more in the scanner). Hence, acquiring images from one subject 3 times and treating those values as independent can bias the measurement. Any data from a single subject should be averaged/combined into a single point before statistical consideration (e.g. ANOVA, binomial test). Similarly, when constructing the cases for review by the 10 radiologists, the number of slices included from a given volunteer should be equal. Ultimately, in most of these analyses there are only 13 subjects, not 19.

2) Matched pairs and image quality: Please clarify in the text if the “slice pairs” shown to radiologists for evaluation where of the same slice on the same subject. This is implied in the text but not made explicit. If it is the case, then the analysis made sense. If it isn’t then the comparison analysis does not make sense at all and is probably the incorrect approach. Also, please clarify why there is no pathway for the radiologist to grade images in terms of quality? T

The current analysis allows an observer to pick one or the other image (or both), but it does not assess how many times the reconstruction produced images that were inadequate for interventional guidance either due to incoherent data or residual image artifact. That result is simply hidden by the observer choosing the alternative (especially since there is no “neither” option). In other words, though proposed technique may receive more “points” that the reference technique, if it fails to reconstruct an adequate image within the 4D volume, then it may not be worth pursuing. To claim superiority, the authors need to demonstrate that the images that are reconstructed are not junk any more than those reconstructed in the original method.

3) Scan time: The acquisition time for the proposed method is clearly an issue. I recommend the authors discuss potential techniques for speeding up acquisition. I would suggest that the authors should include the use of phased array coils which would allow for the use of parallel imaging for in plane acceleration, or even more aptly suited, the use of SMS (simultaneous multislice imaging) which is ideal for this approach. I have yet to see a single interventional MRI procedure in which phased array coils were not used during pre-procedural imaging and I believe that in an effort to increase breadth of applicability by relying only on the body coil for signal, the authors are hurting their actual applicability due prohibitively long scan times.

Specific Comments

Page 3/13, Methods, first paragraph: “TR = 40.0 ms” is manufacturer specific term since no modern TRUFI sequence can run with this TR successfully (especially at 3T). The parameters that should be reported in this section but aren’t include: (1) real repetition time (TR) which in Siemens-speak is referred to as echo spacing (likely 2*TE), (2) the actual number of ky-lines acquired within the matrix of 140 phase encodes and descriptions of whether partial Fourier sampling in ky (or assymetric echo in kx) is applied, (3) the readout bandwidth, (4) the min,max and average number of slices acquired per subject (assume 38-57 but average should be reported). In the current description, if 140 lines of k-space were acquired in 200 ms (ignoring start/stop subsequences), then each line would take 1.43 ms, not a feasible number given the matrix size. Hence, the authors need to specify the acquisition more correctly to ensure reproducibility for the reader to understand the current acquisition strategy since it has great impact on temporal resolution.

Page 3/13, Methods, first paragraph: “… body coil…” Clarifying terminology about body coils as there may be some confusion here. The term “body coil” is used to describe the large-bore fixed birdcage coil used for homogeneous RF transmission, rarely used for imaging in diagnostic MRI due to extremely low SNR. A body array (a coil composed of multiple elements) represents the standard surface coils arrays with multiple receive channels that are standard in diagnostic MRI. It is not clear which was used for imaging here, though by the description the authors used no body arrays and used only the body coil for imaging. If this is correct, please clarify the terminology.

Page 4/13: “localisators” is not MR terminology or standard English. The typically used term is "localizer(s)" or "localiser(s)".

Page 6/13: The phrase "well trackable" is awkward in English. Maybe "easily trackable" or "robustly trackable"? Also, since the term is inherently subjective, it may be helpful to display a figure/panel with several patterns that do meet the criteria and maybe even some patterns that do not meet the desired criteria. This is central to the paper as in the end, the authors are using cross correlation on a vessel pattern to estimate respiratory state.

Figure 10: There should be error bars for each point? If my understanding is correct, each point represents averages across all subjects?

7. PLOS authors have the option to publish the peer review history of their article (what does this mean?). If published, this will include your full peer review and any attached files.

Reviewer #2: No

Reviewer #3: No

---

## [Author Response · Author response to Decision Letter 1]

20 May 2020

Reviewer 3 02.: The Methods section clarifies the method, but it is not clear where the original ends and where the new one begins. This should be crystal clear for any reader without having to read the entirety of the work.

> You are right, we were not absolutely clear regarding where the baseline method ends and our method starts. We made changes to the manuscript to clarify which sections describe the general concept and common aspects that our method shares with the baseline method. We also made clear which sections describe the parts that differentiate our method from the baseline. (cf. page 3 lines 71 – 74 in the revised manuscript with track changes) 

Reviewer 3 03.: Any data from a single subject should be averaged/combined into a single point before statistical consideration (e.g. ANOVA, binomial test). Similarly, when constructing the cases for review by the 10 radiologists, the number of slices included from a given volunteer should be equal.

> Thank you for pointing out the possibility of a bias in our analysis. We made changes to the statistical analysis. We now average data points of the same subject and made changes to the manuscript to reflect that point. After the change, all reported main and interaction effects are present. (cf. page 7 lines 224 – 225 and page 11 table and page 12 table, in the revised manuscript with track changes) Please note that due to the annotation the table does not fit on the page. Please also refer to the manuscript without tracked changes.

Reviewer 3 04.: Please clarify in the text if the “slice pairs” shown to radiologists for evaluation where of the same slice on the same subject.

> Thank you for pointing that out. We made changes to the manuscript to clarify that slices of a pair were of the same subject, position and orientation. (cf. page 8 lines 250 – 254 in the revised manuscript with track changes) 

Reviewer 3 05.: Also, please clarify why there is no pathway for the radiologist to grade images in terms of quality? The current analysis allows an observer to pick one or the other image (or both), but it does not assess how many times the reconstruction produced images that were inadequate for interventional guidance either due to incoherent data or residual image artifact. That result is simply hidden by the observer choosing the alternative (especially since there is no “neither” option). In other words, though proposed technique may receive more “points” that the reference technique, if it fails to reconstruct an adequate image within the 4D volume, then it may not be worth pursuing. To claim superiority, the authors need to demonstrate that the images that are reconstructed are not junk any more than those reconstructed in the original method.

> We understand that an absolute scale contains more information and is more potent than a mere comparison, which is a valid point. We made changes to the manuscript to make clear that we aim to evaluate whether our methods additions improve the reconstruction quality over the baseline showing that this is a valid path. We also made it clear that from our analysis we cannot and do not derive a conclusion whether the reconstruction quality is good enough for any given use case. (cf. page 8 lines 242 – 243 and page 11 lines 385 – 390 in the revised manuscript with track changes) 

Reviewer 3 06.: Scan time: The acquisition time for the proposed method is clearly an issue. I recommend the authors discuss potential techniques for speeding up acquisition. I would suggest that the authors should include the use of phased array coils which would allow for the use of parallel imaging for in plane acceleration, or even more aptly suited, the use of SMS (simultaneous multislice imaging) which is ideal for this approach. I have yet to see a single interventional MRI procedure in which phased array coils were not used during pre-procedural imaging and I believe that in an effort to increase breadth of applicability by relying only on the body coil for signal, the authors are hurting their actual applicability due prohibitively long scan times.

> You noticed that phased array coils allow for SMS with faster imaging and better SNR. We agree that this in an improvement and have made changes to the manuscript to discuss the possibilities and how our method would integrate with that. (cf. page 10 lines 337 – 352 in the revised manuscript with track changes)

Reviewer 3 07.: Page 3/13, Methods, first paragraph: “TR = 40.0 ms” is manufacturer specific term since no modern TRUFI sequence can run with this TR successfully (especially at 3T). The parameters that should be reported in this section but aren’t include: (1) real repetition time (TR) which in Siemens-speak is referred to as echo spacing (likely 2*TE), (2) the actual number of ky-lines acquired within the matrix of 140 phase encodes and descriptions of whether partial Fourier sampling in ky (or assymetric echo in kx) is applied, (3) the readout bandwidth, (4) the min,max and average number of slices acquired per subject (assume 38-57 but average should be reported). In the current description, if 140 lines of k-space were acquired in 200 ms (ignoring start/stop subsequences), then each line would take 1.43 ms, not a feasible number given the matrix size. Hence, the authors need to specify the acquisition more correctly to ensure reproducibility for the reader to understand the current acquisition strategy since it has great impact on temporal resolution. 

> Thank you for pointing that out. We made changes to the manuscript to add this information. (cf. page 3 lines 77 – 83, page 4 line 120 in the revised manuscript with track changes)

Reviewer 3 08.: Page 3/13, Methods, first paragraph: “… body coil…” Clarifying terminology about body coils as there may be some confusion here. The term “body coil” is used to describe the large-bore fixed birdcage coil used for homogeneous RF transmission, rarely used for imaging in diagnostic MRI due to extremely low SNR. A body array (a coil composed of multiple elements) represents the standard surface coils arrays with multiple receive channels that are standard in diagnostic MRI. It is not clear which was used for imaging here, though by the description the authors used no body arrays and used only the body coil for imaging. If this is correct, please clarify the terminology.

> Thank you for pointing that out. We made changes to the manuscript to clarify the terminology. (cf. page 3 lines 87 – 88 in the revised manuscript with track changes)

Reviewer 3 09.: Page 4/13: “localisators” is not MR terminology or standard English. The typically used term is "localizer(s)" or "localiser(s)".

> Thank you for pointing that out. We corrected this in the manuscript (cf. page 4 lines 122 and 126 in the revised manuscript with track changes)

Reviewer 3 10.: Page 6/13: The phrase "well trackable" is awkward in English. Maybe "easily trackable" or "robustly trackable"? Also, since the term is inherently subjective, it may be helpful to display a figure/panel with several patterns that do meet the criteria and maybe even some patterns that do not meet the desired criteria. This is central to the paper as in the end, the authors are using cross correlation on a vessel pattern to estimate respiratory state.

> Thank you for pointing that out. We made changes to the manuscript to clarify that point. (cf. page 6 lines 175 – 181 in the revised manuscript with track changes) 

Reviewer 3 11.: Figure 10: There should be error bars for each point? If my understanding is correct, each point represents averages across all subjects?

> Thank you for this point. We added error bars to represent the standard deviation and made changes to the manuscript to reflect that point. (cf. page 10 Fig. 10 in the revised manuscript)

---

## [Decision Letter · Decision Letter 2]

10 Jun 2020

4D MRI: Robust sorting of free breathing MRI slices for use in interventional settings

PONE-D-19-18726R2

Dear Dr. Gulamhussene,

We’re pleased to inform you that your manuscript has been judged scientifically suitable for publication and will be formally accepted for publication once it meets all outstanding technical requirements.

Kind regards,

Haydar Celik, PhD

Academic Editor

PLOS ONE

Additional Editor Comments (optional):

Reviewers' comments:

Reviewer's Responses to Questions

**Comments to the Author**

1. If the authors have adequately addressed your comments raised in a previous round of review and you feel that this manuscript is now acceptable for publication, you may indicate that here to bypass the “Comments to the Author” section, enter your conflict of interest statement in the “Confidential to Editor” section, and submit your "Accept" recommendation.

Reviewer #2: All comments have been addressed

Reviewer #3: All comments have been addressed

2. Is the manuscript technically sound, and do the data support the conclusions?

Reviewer #2: Yes

Reviewer #3: Yes

3. Has the statistical analysis been performed appropriately and rigorously? 

Reviewer #2: Yes

Reviewer #3: Yes

4. Have the authors made all data underlying the findings in their manuscript fully available?

Reviewer #2: Yes

Reviewer #3: Yes

5. Is the manuscript presented in an intelligible fashion and written in standard English?

Reviewer #2: Yes

Reviewer #3: Yes

6. Review Comments to the Author

Reviewer #2: I believe the authors have addressed all of the concerns of the reviewers in their revised manuscript.

Reviewer #3: The majority of my comments have been addressed appropriately.

The majority of my comments have been addressed appropriately.

7. PLOS authors have the option to publish the peer review history of their article (what does this mean?). If published, this will include your full peer review and any attached files.

Reviewer #2: No

Reviewer #3: No

---

## [Editor Report · Acceptance letter]

12 Jun 2020

PONE-D-19-18726R2 

4D MRI: Robust sorting of free breathing MRI slices for use in interventional settings 

Dear Dr. Gulamhussene:

I'm pleased to inform you that your manuscript has been deemed suitable for publication in PLOS ONE. Congratulations! Your manuscript is now with our production department. 

Kind regards, 

on behalf of

Dr. Haydar Celik 

Academic Editor

PLOS ONE